# Image Compression with Product Quantized Masked Image Modeling

**Alaaeldin El-Nouby**[◇,§,†]        *aelnouby@meta.com*
**Matthew Muckley**[◇]        *mmuckley@meta.com*
**Karen Ullrich**[◇]        *karenu@meta.com*
**Ivan Laptev**[§,†]        *ivan.laptev@inria.fr*
**Jakob Verbeek**[◇]        *jjverbeek@meta.com*
**Hervé Jégou**[◇]        *rvj@meta.com*

◇*Meta AI, FAIR Team, Paris,* §*ENS, PSL University,* †*INRIA, Paris*

**Reviewed on OpenReview:** *https://openreview.net/forum?id=Z2L5d9ay4B*

## Abstract

Recent neural compression methods have been based on the popular hyperprior framework. It relies on Scalar Quantization and offers a very strong compression performance. This contrasts from recent advances in image generation and representation learning, where Vector Quantization is more commonly employed.

In this work, we attempt to bring these lines of research closer by revisiting vector quantization for image compression. We build upon the VQ-VAE framework and introduce several modifications. First, we replace the vanilla vector quantizer by a product quantizer. This intermediate solution between vector and scalar quantization allows for a much wider set of rate-distortion points: It implicitly defines high-quality quantizers that would otherwise require intractably large codebooks. Second, inspired by the success of Masked Image Modeling (MIM) in the context of self-supervised learning and generative image models, we propose a novel conditional entropy model which improves entropy coding by modelling the co-dependencies of the quantized latent codes. The resulting PQ-MIM model is surprisingly effective: its compression performance is on par with recent hyperprior methods. It also outperforms HiFiC in terms of FID and KID metrics when optimized with perceptual losses (e.g. adversarial). Finally, since PQ-MIM is compatible with image generation frameworks, we show qualitatively that it can operate under a hybrid mode between compression and generation, with no further training or finetuning. As a result, we explore the extreme compression regime where an image is compressed into 200 bytes, i.e., less than a tweet.

## 1 Introduction

Efficient image codecs have accelerated the rapid growth of the internet by enabling the transmission of images in a few dozens of kilobytes, thanks to the emergence of effective lossy methods. This democratization was accompanied by standardization efforts to facilitate the interoperability, which led to the emergence of standards such as the Joint Photographic Experts Groups (JPEG). Subsequent formats have leveraged scientific advances on all components of source coding, ranging from transforms (Antonini et al., 1992), and quantization (Gray & Neuhoff, 1998), to entropy coding (Witten et al., 1987; Taubman, 2000), eventually leading to modern video compression codecs enabling streaming and video-conferencing applications.

Neural methods have recently become increasingly popular for image compression as well as other image processing tasks, such as denoising (Tian et al., 2020), super-resolution (Bruna et al., 2016; Dong et al., 2015; Ledig et al., 2017; Wang et al., 2021) or image reconstruction (Wang et al., 2020; Knoll et al., 2020). In typical scenarios, neural image compression is not necessarily mature enough to take over standard techniques like the BPG format inherited from the High-Efficiency Video Coding standard (Sullivan et al., 2012). This

is because they do not offer a significant quantitative advantage over prior works that would justify the higher complexity, which depends on the context and operational constraints. A key advantage of neural compression methods is their enhanced qualitative reconstruction when incorporating an adversarial loss or likewise psycho-visual objectives favoring visually appealing reconstructions (Agustsson et al., 2019; Mentzer et al., 2020). From this perspective, neural compression is related to image generation. The two subfields, however, are currently dominated by different approaches, noticeably they employ different discretization procedures. Indeed, while earlier neural compression methods utilized vector quantization (Agustsson et al., 2017, VQ), recent methods mostly employ scalar quantization (SQ). In contrast, the recent literature on image generation (Chang et al., 2022; Yu et al., 2021; Esser et al., 2021; Rombach et al., 2022) relies on Vector Quantization jointly with a distortion criterion akin to those used in compression.

In this work we aim to reduce the methodological gap and to make a step towards unification of neural image compression and image generation, and allowing image compression to more directly benefit from the rapid advances in image generation methods. Patch-based masking methods for self-supervised learning (Bao et al., 2022; He et al., 2021b; El-Nouby et al., 2021a) have recently demonstrated their potential for image generation (Chang et al., 2022). Inspired by this work we propose a compression approach built upon Vector Quantized Variational Auto-Encoders (Oord et al., 2017; Razavi et al., 2019). In this context, we focus on two intertwined questions: (1) How to define a vector quantizer offering a range of rate-distortion operating points? (2) How to define an entropy model minimizing the cost of storing the quantization indexes, while avoiding the prohibitive complexity of an auto-regressive model?

To address the challenges above, we revisit vector quantization in image compression, and investigate product quantization Jégou et al. (2010) (PQ) in a compression system derived from VQ-VAE Oord et al. (2016b). We show that PQ offers a strong and scalable rate-distortion trade-off. We then we focus on the spatial entropy modeling and coding of the quantization indexes in the VQ or PQ latent layer, hence we name our method as (Vector/Product)-Quantized Masked Image Modeling (VQ-MIM and PQ-MIM). To this end, we introduce a multi-stage vector-quantized image model: we gradually reduce the conditional entropy of the patch latent codes by increasing the number of observed patches we condition on for each stage. The conditional distribution over patches is estimated by a transformer model and provided to an entropy coder, symmetrically on the emitter and receiver sides.

In summary, we make the following contributions:

- We introduce a novel Masked Image Modeling conditional entropy model that significantly reduces the rates by leveraging the spatial inter-dependencies between latent codes.

- We introduce product quantization for VQ-VAE. This simple PQ-VAE variant offers a strong and scalable rate-distortion trade-off.

- When trained with adversarial and perpetual losses, PQ-MIM exhibits a strong performance in terms of perceptual metrics like FID and KID, outperforming HiFiC (Mentzer et al., 2020).

- We qualitatively show that PQ-MIM is capable of operating in a hybrid mode, between generative and compression, without requiring further training and finetuning. This allows for higher resilience to corrupted or missing signal where our model can fill-in the missing information.

## 2 Related work

**Neural image compression** Early approaches to neural image compression reach back to the late 1980s (Sonehara, 1989; Sicuranza et al., 1990; Bottou et al., 1998). Recent rapid advances in explicit and implicit density modelling (Goodfellow et al., 2014; Kingma & Welling, 2014; Larochelle & Murray, 2011; Oord et al., 2016a; Rezende & Mohamed, 2015; Salimans et al., 2017) have renewed interest in posing image compression as a learning problem. Due to the connection to variational learning (Gregor et al., 2016; Frey, 1998; Alemi et al., 2018), variational auto-encoders have been the primary choice for lossy image compression. In contrast to standard variational models, the evaluation of neural compression models focuses on achievable bitrates, multi-scale applicability and computational complexity. The initial works in the field used fully

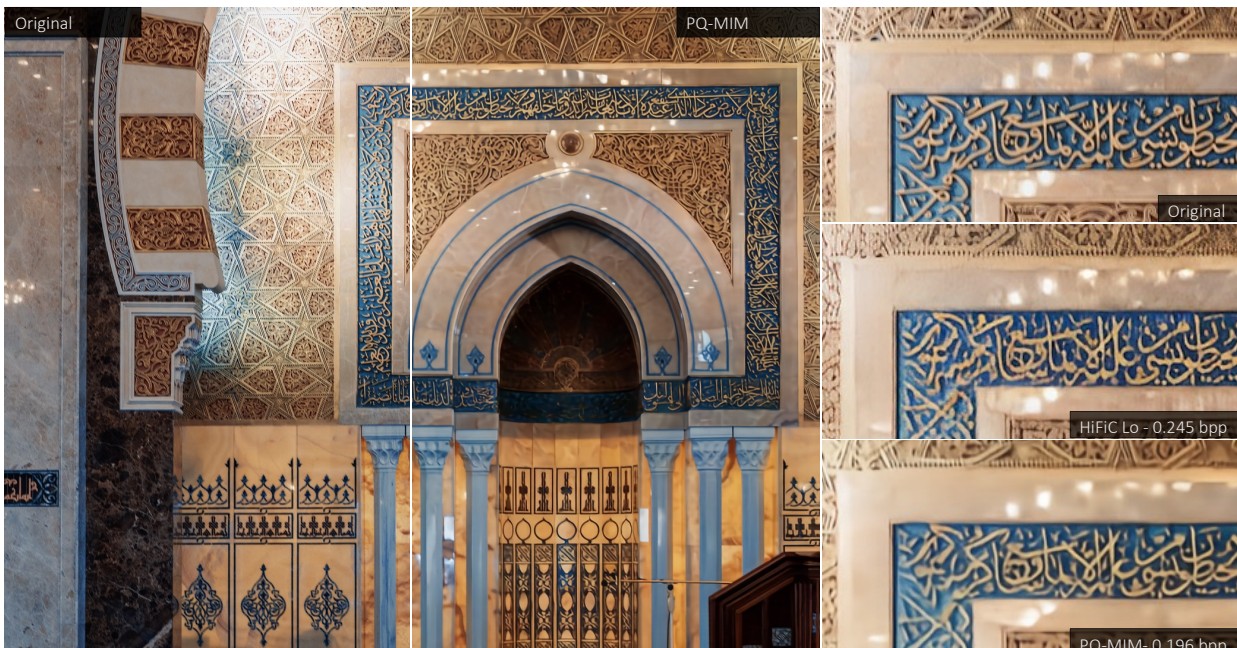

Figure 1: **Qualitative example of PQ-MIM compression.** PQ-MIM provides a strong compression performance. We retain many of the details present original image with minimal blurring effect even with comrpession rate as low as 0.196 bpp. Moreover, compared to HiFiC we achieve a lower rate for the same image. PQ-MIM provides colors that are more faithful to the original image, while HiFiC has a darkening effect and some high frequency artifacts. On the other hand, PQ-MIM can have some smoothing effect that can cause loss of detail for particular regions (e.g. some of the Arabic letters in the example above). More qualitative examples are provided in Appendix D.

convolutional architectures for encoding/decoding (Ballé et al., 2017; Theis et al., 2017; Mentzer et al., 2018). The resulting encoded latent image representations are quantized and compressed via an entropy coder with learned explicit density model or "entropy bottleneck". Initial variational approaches directly modeled a single level of code densities. Ballé extended these models by introducing a second "hyperprior" that yielded improved performance (Ballé et al., 2018). Hyperprior models have been the basis for several subsequent advances with further improvements for density modeling, such as joint autoregressive models (Minnen et al., 2018), Gaussian mixture/attention (Cheng et al., 2020), and channel-wise auto-regressive models (Minnen & Singh, 2020). He et al. (2021a) improve the efficiency of spatial autoregressive context models by utilizing checkerboard contexts which allows for more parallelization friendly decoding. This approach can be considered a special case when we use only two steps of quincunx pattern encoding/decoding. Another line of work has proposed to use vector quantization with histogram-based probabilities for image compression (Agustsson et al., 2017; Lu et al., 2019). Contrary to VQ-VAE models, these models typically optimize the rate (or a surrogate of the rate) directly and may include a spatial component for the quantized vectors. Yang et al. (Yang et al., 2020) showed multiple ways to improve the encoding process, including the fine-tuning of the discretization process and employing bits-back coding in the entropy bottleneck. Finally, (Santurkar et al., 2018; Mentzer et al., 2020; Rippel & Bourdev, 2017; Agustsson et al., 2019) showed that altering the distortion metric to include an additional adversarial loss can make a large difference for compression rate. Another interesting line of work considers image compression by training image-specific networks, or network adapters, that map image coordinates to RGB values, and compressing the image-specific parameters (Dupont et al., 2021; 2022; Strüpler et al., 2022).

**VQ-models for image generation.**     There has been significant interest in generative models based in discrete image representations, as introduced by VQ-VAE (Oord et al., 2017; Razavi et al., 2019). A discrete representation of reduced spatial resolution is learned by means of an autoencoder which quantizes the latent representations. This discrete representation is coupled with a strong prior, for example implemented as an autoregressive pixel-CNN model. VQ-GAN (Esser et al., 2021) replaces the prior architecture with a transformer model (Vaswani et al., 2017), and introduces an adversarial loss term to learn an autoencoder with more visually pleasing reconstructions and improved sample quality. This approach has been extended to text-based generative image models by extending the prior to model a longer sequence that combines the discrete

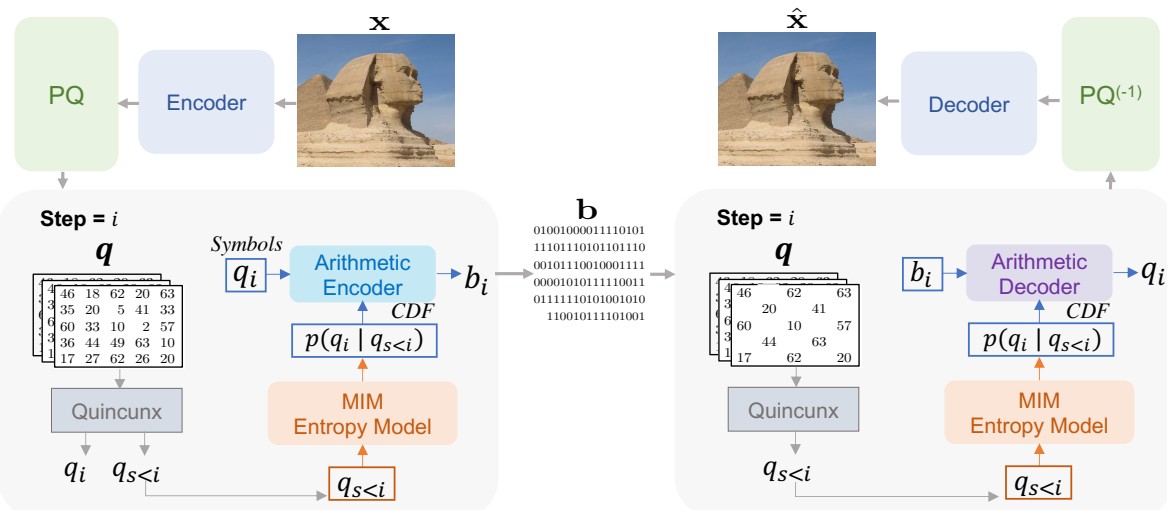

Figure 2: **PQ-MIM overview.** Our model consists of (i) a a transformer based encoder and decoder, (ii) a masked image model (MIM) for conditional entropy modeling, and (iii) an entropy coder, *e.g* an arithmetic coder (AE/AD). The input image **x** is projected to a set of latent features, followed by product quantization to yield quantization indices **q**. The arithmetic coder encodes (and decodes) **q** into a bitstream **b** in a lossless manner. The elements in **q** are spatially split into groups, as detailed in Figure 4. **Conditional Entropy Modeling.** Our model estimates the conditional probabilities of the discrete indices in $\mathcal{S}$ steps. Every step, a subset of the tokens $q_i$ is selected using the quincunx pattern. Our MIM transformer estimates $p(q_i|q_{s<i})$ and passes it to the Arithmetic encoder as a CDF, effectively reducing the lossless compression cost.

image representation with a prefix that encodes the conditioning text. This has yielded impressive results by scaling the model capacity and training data to tens or hundreds of million text-image pairs (Ding et al., 2021; Gafni et al., 2022; Ramesh et al., 2021). A fundamental limitation of autoregressive generative models is that they sample data sequentially, requiring separate non-parallel evaluation of the predictive (transformer) model to sample each token. To alleviate this, several data items can be sampled independently in parallel, conditioning on all previously sampled tokens. This has been leveraged to speed-up pixel-CNNs for small images and video by two to three orders of magnitude (Reed et al., 2017). More recently, MaskGIT (Chang et al., 2022) and follow-up work (Lezama et al., 2022) explored this for generative models of VQ-VAE representations, and find that similar or better sample quality is obtained by parallel sampling of image patch subsets in few steps, reducing the generation time significantly. Despite their success for image synthesis we are not aware of the use of such models for image compression in earlier work.

## 3    Product Quantized Masked Image Modeling

This work takes a step towards to closing the gap between neural image compression and image generation methodologies. We revisit vector quantization for image compression and propose an entropy model inspired by masked image modelling. Our compression pipeline, depicted in Figure 2, relies on three neural networks:

1. **The Encoder network** $\hat{E} : \mathcal{X} \to \mathcal{Z}$ maps input images $\mathbf{x} \in \mathcal{X}$ to a quantized representation $\mathbf{z} \in \mathcal{Z}$.

2. **The Mask Image Model** (MIM) compresses the quantized representations without loss of information. This network is involved both on the compression and decompression side.

3. **The Decoder network** $G : \mathcal{Z} \to \mathcal{X}$ produces an estimate $\hat{\mathbf{x}} = G(\hat{E}(\mathbf{x}))$ of the original image $\mathbf{x}$.

We now detail the architecture, in particular our PQ proposal, of the image model that we employ in the statistical lossless coding, as well as the training scheme.

### 3.1   High-level architecture: PQ-VAE

**High-level architecture.**    We follow recent work on discrete generative image models for the design of image encoder and decoder (Chang et al., 2022; Esser et al., 2021; Oord et al., 2017; Razavi et al., 2019; Yu

et al., 2021). The encoder $E : \mathcal{X} \to \mathbb{R}^{w \times h \times d}$ takes an RGB image $\mathbf{x}$ of resolution $W \times H$ as input and maps it to a latent representation $E(\mathbf{x})$ with $d$ feature channels and a reduced spatial resolution $w \times h$, downsampling the input resolution by a factor $f = W/w = H/h$. The $T = w \times h$ elements of the latent representation $E(\mathbf{x})$ are quantized with a vector quantizer $Q(\cdot)$ to produce the quantized latent representation $\mathbf{z} = Q(E(\mathbf{x})) = \hat{E}(\mathbf{x})$, where each element in $E(\mathbf{x})$ is replaced with its nearest cluster center. The decoder $G$ uses the quantized latents $\mathbf{z}$ to reconstruct the image.

**Product Quantization.** In VQ-VAE, the quantizer $Q$ is simply an online k-means quantizer that produces quantization indices from real-valued vectors. We denote by $\mathbf{q} \in \{1, \dots, V\}^T$ the map of quantization indices indicating for each element of $\mathbf{z}$ which of the $V$ centroids is selected. The higher $V$ the more precise is the approximation $\mathbf{z}$, leading to higher bit rates. For instance, assuming that indices are coded with a naive coding scheme (see next section), the bit rate is doubled when moving to $V = 256$ to $V = 65536$ centroids.

However, scaling the number of centroids $K$ is possible up to thousands of centroids, but beyond that it is computationally prohibitive. Additionally, it is challenging to train large codebooks where each centroid has a very low probability of being updated. To address this problem, we replace the online k-means quantizer by a product quantizer (Jégou et al., 2010) (PQ): the latent vector $\mathbf{z}$ is split into $M$ subvectors as $\mathbf{z} = [\mathbf{z}^1, \dots, \mathbf{z}^j, \dots, \mathbf{z}^M]$ of dimension $M/d$. Each subvector is quantized by a distinct quantizer having $V_s$ quantization value. The set of quantizers implicitly defines a vector quantizer in the latent space with $V = V_s^M$ distinct centroids. Hence, we can easily define very large codebooks without the computational and optimization problems mentioned above, because both the assignment and learning are marginalized over the different subspaces. Empirically, we observe in Figure 3 that PQ provides a better scaling behaviour for higher rates compared to VQ whose codebook size needs to grow exponentially to achieve the same rates.

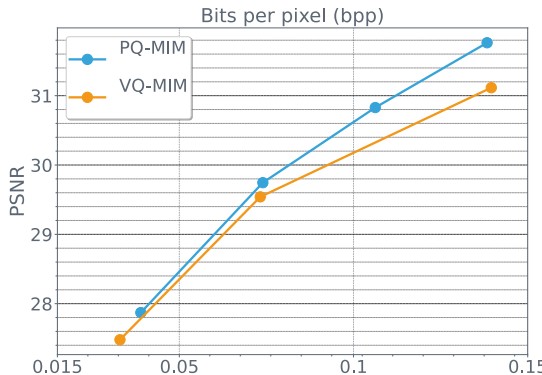

Figure 3: **PQ and VQ comparison**. While VQ provides a comparable performance to PQ for extremely low rates where the codebook size is small, PQ exhibits better scaling behaviour for higher rates. For VQ we only vary the codebook size, while for PQ we fix the codebook size to 256 and vary the number of sub-vectors between 2 and 6.

**Neural network.** Without loss of generality, we choose all neural network models to be identical. This is not a requirement but this offers the property that the encoder and decoder have identical complexities, and that the memory and compute peaks are identical. More specifically we choose a cross-variance transformer (El-Nouby et al., 2021b) (XCiT), whose complexity is linear with respect to image resolution. In contrast, standard vision transformers (Dosovitskiy et al., 2021) (ViT) are quadratic in the image surface, which is prohibitive for high resolution images that can typically require strong compression. We point out that recent work has shown that Swin-Transformers (Zhu et al., 2021) could be a compelling choice as well in the context of image compression. Formally they also have a quadratic complexity, but this is amortized by the hierarchical structure of this architecture.

### 3.2 Image entropy model

In this section we present PQ-MIM. The objective is to compress the discrete representations $\mathbf{q}$ without loss of information, producing a bitstream of the compressed representation that can be transmitted or stored. During the decoding stage, we invert the aforementioned lossless compression. This model can be regarded as the VQ/PQ-VAE counterpart of adaptive contextual arithmetic coders, like EBCOT (Taubman, 2000) or CABAC (Richardson, 2004), proposed in early compression standards, in that it couples a conditional probabilistic model with an arithmetic coder.

**Lossless compression.** A naive manner for lossless compression of the discrete image codes $\mathbf{q} = \{q_t\}_{t=1}^T$ is to use fixed-length codes. In that case, each code word is assigned to a unique binary representation of equal length, resulting in $\lceil \log_2 V \rceil$ bits per element $q_t$. This approach is computationally very efficient as fixed-length codes are not model-based, and as such do not require likelihood estimation, and because all

$$p(q_1) \qquad p(q_2|q_1) \qquad p(q_3|q_2,q_1) \qquad p(q_4|q_3,q_2,q_1) \qquad p(q_5|q_4,q_3,q_2,q_1)$$

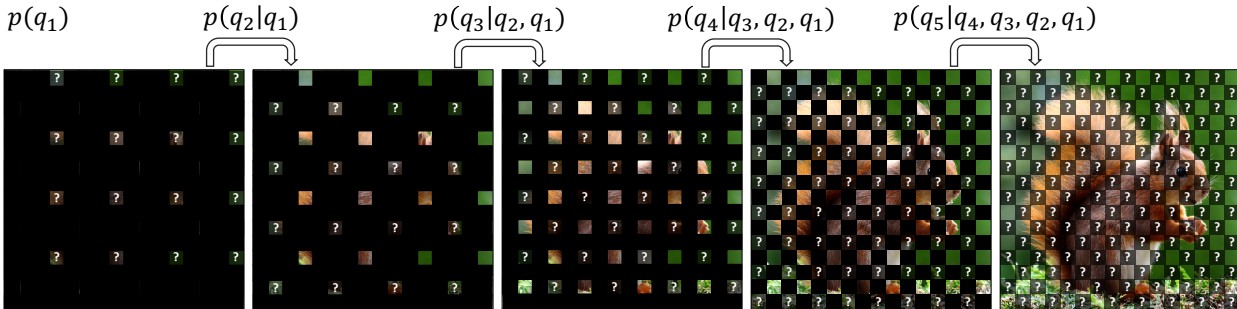

Figure 4: **Illustration of PQ-MIM with a quincunx pattern.** We employ the quincunx pattern both on the encoder and decoder side. Each panel represents one of five stages in which we en/de-code a set of tokens in parallel using a probability model $p_s$ implemented by a transformer with parameters $\theta_s$. The transformer predicts the tokens in $\mathbf{q}_s$, displayed in grayscale and marked by "**?**", and takes as input the preceding groups of tokens $\mathbf{q}_1, \ldots, \mathbf{q}_{s-1}$ that are displayed in color. The distribution provided by this neural network is fed to an arithmetic en/de-coder.

codes are of equal length by construction, the computation is perfectly parallelizable. However, theoretically this coding scheme could be Shannon optimal only if codewords are uniformly and independently distributed. Those assumptions are not met in practice due to the architecture choices we have made previously: k-means does not produce uniformly distributed indices except in singular cases (Gray & Neuhoff, 1998). More details about lossless compression are covered in Appendix A.

**Entropy model.**   To improve the bitrate, we hence rely on an entropy coder, which provides an inverse pair of functions, $\texttt{enc}_p$ and $\texttt{dec}_p$, achieving near optimal compression rates on sequences of symbols for any distribution $p$. The better $p$ matches the (unknown) underlying data distribution, the better the compression rate (Cover, 1991). Generally, more powerful generative models will ensure better compression performance.

Fully autoregressive generative models $p(\mathbf{q}) = \prod_{t=1}^{T} p(q_t|\mathbf{q}_{<t})$ are powerful (Ding et al., 2021; Esser et al., 2021; Gafni et al., 2022; Ramesh et al., 2021; Yu et al., 2021), however, they are inconvenient in that the likelihood estimation for this type of model is not trivially parallelizable: each patch index must be processed sequentially as it is used to condition subsequent patch indices. Thus, similar to prior works (Chang et al., 2022; Reed et al., 2017) we propose a masked image model, which we use to predict the image patch indices in several stages. Specifically, we partition $\mathbf{q}$ into $S$ subsets $\mathbf{q}_1, \mathbf{q}_2, \ldots, \mathbf{q}_S$ of patch indices, that we refer to as *tokens* by analogy to language modelling:

$$\mathbf{q} = \bigcup_{s=1}^{S} \mathbf{q}_s. \tag{1}$$

We model the elements in each subset conditionally independent given all preceding groups:

$$p(\mathbf{q}) = \prod_{s=1}^{S} p\left(\mathbf{q}_s | \mathbf{q}_{<s}; \theta_s\right), \tag{2}$$

$$p\left(\mathbf{q}_s | \mathbf{q}_{<s}; \theta_s\right) = \prod_{q_t \in \mathbf{q}_s} p\left(q_t | \mathbf{q}_1, \mathbf{q}_2, \ldots, \mathbf{q}_{s-1}; \theta_s\right). \tag{3}$$

Since $p(\mathbf{q}_1)$ is not conditioned on any previous elements, it fully factorizes over $q_t \in \mathbf{q}_1$, and we model it as the marginal distribution over the vocabulary observed on the training data. The non-trivial conditional distributions $p(\mathbf{q}_s|\mathbf{q}_{<s}; \theta_s)$ for $s \geq 2$ are modeled using transformer networks, which have few inductive biases and have been successful across many tasks, including image generation. An overview of the MIM entropy model is illustrated in Figure 2.

**Amortized encoding and decoding.**   From a computational point of view, our proposal allows for the compression (resp. decompression) to proceed in $S$ stages. In each stage $s$ we encode/decode the set $\mathbf{q}_s$ of tokens conditioned on the groups $\mathbf{q}_1, \ldots, \mathbf{q}_{s-1}$ encoded/decoded in preceding stages, but independently among the tokens in the set $\mathbf{q}_s$. This allows for paralellization among the elements in each subset $\mathbf{q}_s$, and

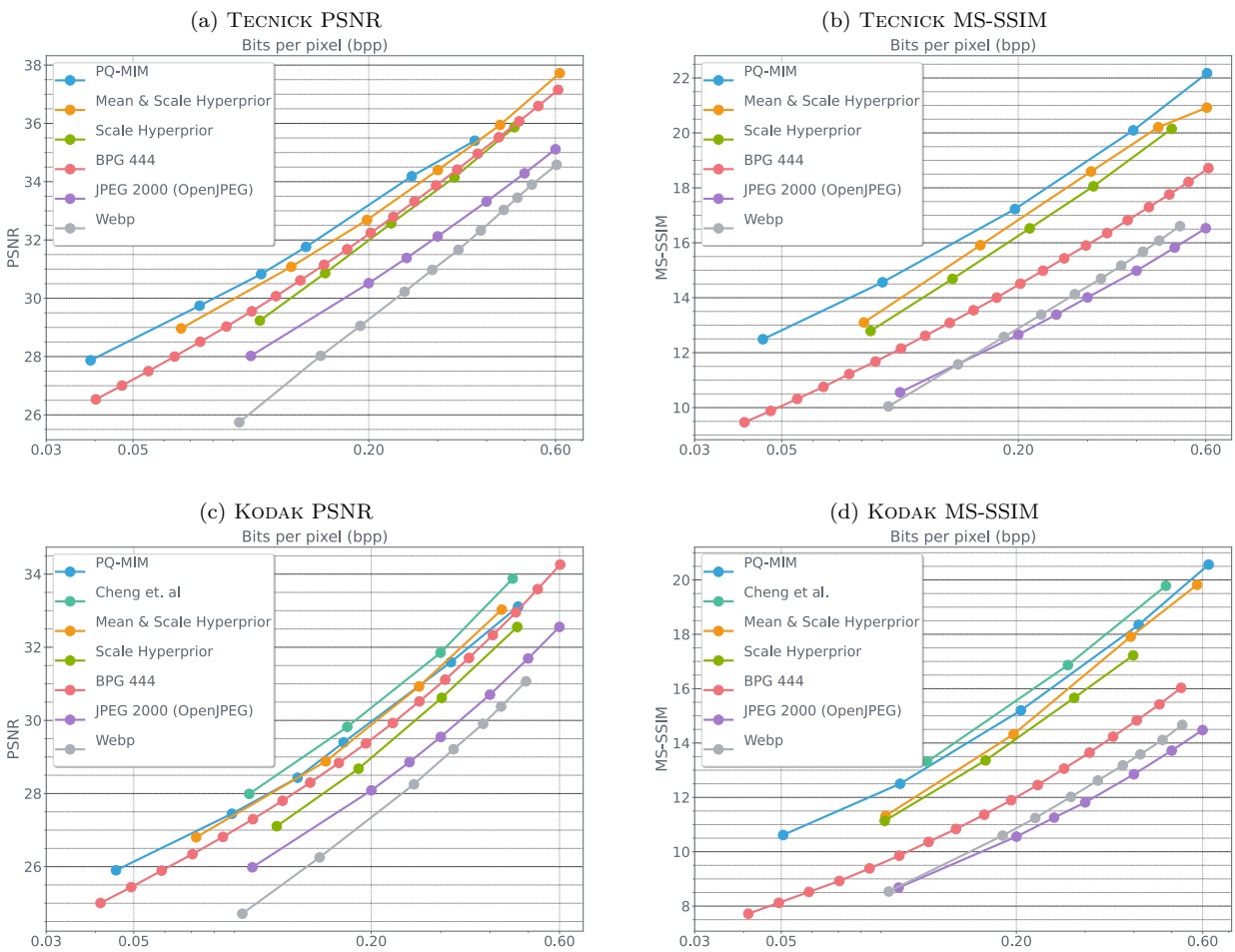

Figure 5: **Rate-Distortion performance for Tecnick and Kodak datasets.** We report PQ-MIM PSNR and MS-SSIM performance for various operating points. PQ-MIM provides a competitive performance, particularly for MS-SSIM, compared to standard codecs such as JPEG 2000 (Taubman & Marcellin, 2012) and BPG (Bellard) as well as recent neural methods (Cheng et al., 2020; Minnen et al., 2018; Ballé et al., 2018).

requires strictly $S$ forward passes through the model independent of the image size, rather than $T$ sequential forwards passes for fully autoregressive models.

**Quincunx partitioning.** In practice, we typically use $S = 5$ stages. We have explored different patterns to partition the $T$ tokens over the $S$ stages. In particular we consider the "quincunx" regular grid pattern, where in each stage we double the number of tokens to predict, see Figure 4 for an illustration. This multi-level refinement was previously explored for image compression in the context of lifting schemes designed with oriented wavelets (Chappelier & Guillemot, 2006). In our experiments we contrast this partitioning with alternative ones with other patterns and subset cardinalities (Figure 9).

### 3.3 Training the PQ-MIM

**Reconstruction objective and training.** The goal of lossy image compression is to match the image and its reconstruction as closely as possible according to some distortion metric. In this paper, we train our model to reduce the image distortion and the quantization loss using the following objective:

$$L = L_{\text{Rec}} + \eta \cdot L_{\text{PQ}} \tag{4}$$

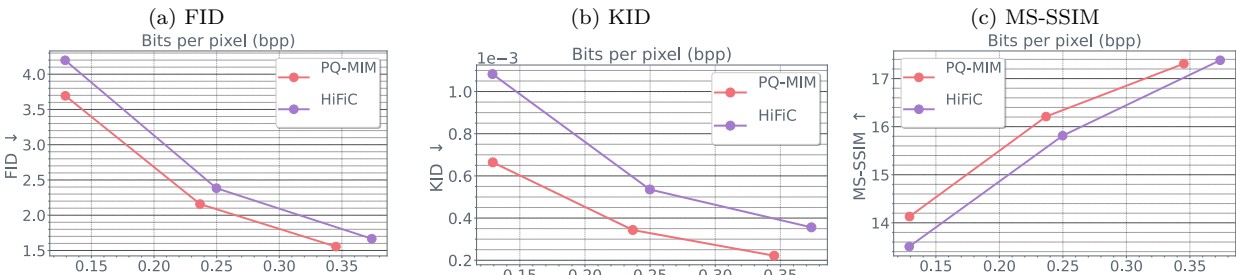

Figure 6: **Perceptual training and evaluation using CLIC 2020 test-set.** Performance of our adversarially trained PQ-MIM w.r.t percpetual metrics compared to HiFiC Mentzer et al. (2020). PQ-MIM provides a stronger performance on FID, KID and MS-SSIM across all reported operating points.

Following VQ-VAE, our quantization objective $L_{\text{PQ}}$ consists of an embedding and commitment losses, averaged over the $M$ different PQ sub-vectors. For the distortion loss $L_{\text{Rec}}$, we present two setups where we use different types of distortion measures:

- **MSE & MS-SSIM.** Typical distortion measures used in the majority of the neural compression literature such as mean squared error (MSE) or multi-scale structural similarity (Wang et al., 2003) (MS-SSIM). For this setup, the model is trained solely using one distortion measure at a time.

$$L_{\text{Rec}}(\mathbf{x}, \hat{E}, G) = L_{\text{MSE/MS-SSIM}}(\mathbf{x}, \hat{\mathbf{x}}) \tag{5}$$

- **Perceptual measures.** Alternatively, we report a setup where we utilize perceptual objectives such as LPIPS (Zhang et al., 2018) and adversarial training (Goodfellow et al., 2014) to enhance psycho-visual image quality. The distortion loss is defined as:

$$L_{\text{Rec}}(\mathbf{x}, \hat{E}, G) = L_{\text{MSE}}(\mathbf{x}, \hat{\mathbf{x}}) + \alpha \cdot L_{\text{Perc}}(\mathbf{x}, \hat{\mathbf{x}}) + \gamma \cdot L_{\text{Adv}}(\hat{E}, G, D), \tag{6}$$

where $\alpha$ and $\gamma$ are weighing coefficients and the adversarial loss $L_{\text{Adv}}$ is defined as:

$$L_{\text{Adv}}(\hat{E}, G, D) = \mathbb{E}_{\mathbf{x}}[\ln D(\mathbf{x})] + \mathbb{E}_{\hat{\mathbf{x}}}[\ln(1 - D(\hat{\mathbf{x}}))], \tag{7}$$

where $D(\cdot)$ is the discriminator and $\mathbb{E}_{\mathbf{x}}$ denotes the expectation over $\mathbf{x}$ sampled uniformly from the training set. Similarly $\mathbb{E}_{\hat{\mathbf{x}}}$ denotes the expectation over reconstructed training images. Note that, unlike the MSE and perceptual losses, the adversarial loss does not compare individual images and their reconstructions, but aims to match the *distributions* of original images and their reconstructions.

**Training the entropy model.** Our MIM module is an XCiT transformer that accepts $T$ tokens as input representing the image patches. During training, we randomly mask a set of tokens by sampling from a uniform distribution $U(0, 1)$. The masked tokens are replaced with a mask embedding vector, while the observed token indices are mapped to their corresponding continuous representation using an embedding look-up table. The MIM module outputs a context vector for every masked token which in turn is passed to $M$ linear heads, representing the different PQ sub-vector indices, followed by a softmax to yield a distribution $p(\boldsymbol{q}_s|\boldsymbol{q}_{<s})$. The module is trained using a standard cross-entropy objective. While we train the autoencoder and the MIM modules simultaneously, we do not backpropagate gradients from the MIM module to the encoder $E$ or the quantization parameters, so the two components can be trained separately in sequence.

## 4 Experiments

We first present our experimental setup in Section 4.1 and the present our results in Section 4.2. We provide ablation studies in Section 4.3. We will share our code and models.

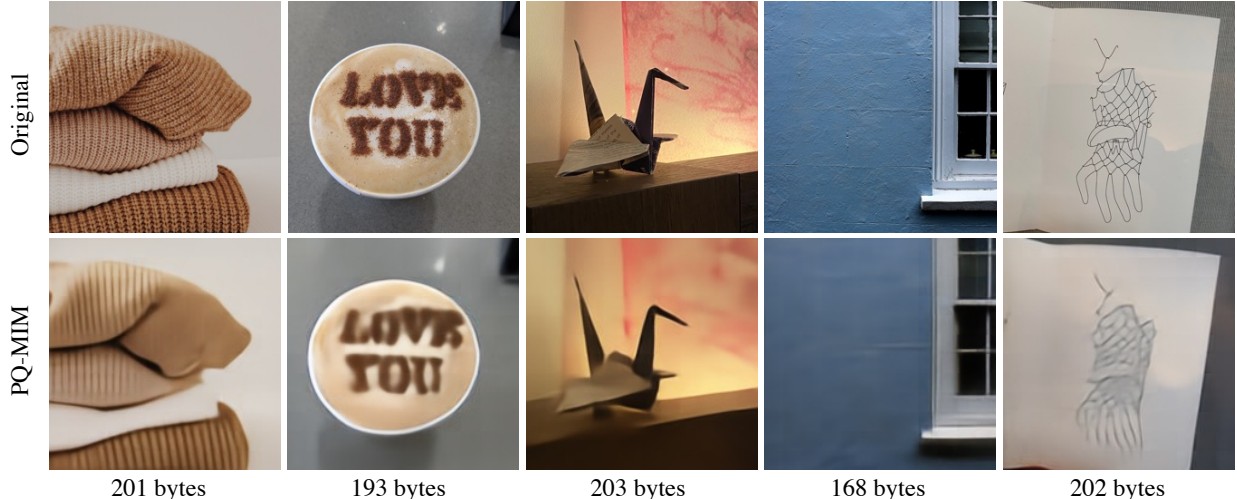

Figure 7: **Extreme Image Compression.** PQ-MIM exhibits non-trivial compression performance at the extreme compression regime (e.g. 0.03 bpp), leading to compressed image codes that can fit in a short tweet (280 characters).

## 4.1 Experimental setup

**Rate-distortion control.** For all our experiments we fix the codebook size $V = 256$ and only vary the number of sub-vectors $M \in \{2, 4, 6\}$ for two different down-sampling factors $f \in \{8, 16\}$.

**PQ-VAE implementation details.** Our PQ-VAE training uses the straight-through estimator (Bengio et al., 2013) to propagate gradients through the quantization bottleneck. As for the quantization, the elements of the latent representation $\mathbf{z}$ are first linearly projected to a low dimensional look-up vector (dim=8 per sub-vector) followed by $\ell_2$ normalization, following Yu et al. (2021). We train our model using ImageNet (Deng et al., 2009) for 50 epochs with a batch size of 256. We use an AdamW (Loshchilov & Hutter, 2019) optimizer with a peak learning rate of $1.10^{-3}$, weight decay of 0.02 and $\beta_2 = 0.95$. We apply a linear warmup for the first 5 epochs of training followed by a cosine decay schedule for the remaining 45 epochs to a minimum learning rate of $5.10^{-5}$. Unless mentioned otherwise, for all experiments, the encoder and decoder use an XCiT-L6 with 6 layers and hidden dimension of 768. We use sinusoidal positional embedding (Vaswani et al., 2017) such that our model can flexibly operate on variable sized images.

For the results reported in Figure 5, the models are trained using solely MSE ($\eta = 0.5$) or MS-SSIM ($\eta = 10.0$) distortion losses for their corresponding plots. As for the models trained with perceptual objectives (Figure 6 and Table 1), they are trained with a weighted sum of MSE, LPIPS ($\alpha = 1$) and adversarial loss ($\gamma = 0.1$). We use a ProjectedGAN Discriminator (Sauer et al., 2021) architecture. The perceptual training is initialized with an MSE only trained checkpoint and trained for 50 epochs using ImageNet with a learning rate of $10^{-4}$ and weight decay of $5.10^{-5}$. Similar to HiFiC (Mentzer et al., 2020), we freeze the encoder during the perceptual training. Additionally, we find that clipping the gradient norm to a maximum value of 4.0 improves the training stability. Our discriminator takes only the decoded image as an input and does not rely on any other conditional signal.

**MIM implementation details.** Our MIM module is an XCiT-L12 with 12 layers and embedding dimension of 768. MIM is trained simultaneously with the PQ-VAE, but the gradients are not backpropagated to the encoder or the quantizer parameters. The PQ indicies $q$ are split into inputs and targets for the MIM model as defined by the quincunx partitioning pattern. By default we use $S = 5$ stages. All stages are processed with the same MIM model. The masked patches are replaced by a learnable "mask" token embedding. The loss is computed only for the masked patches. Since every token is assigned $M$ PQ indices, the output of the MIM transformer is passed to $M$ separate linear heads to predict $M$ softmax normalized distributions over their corresponding codebooks. The marginal distribution of the codebook is computed as a normalized histogram over the ImageNet training set. For entropy coding, we use the implementation of the `torchac`[1] arithmetic coder.

---

[1] https://github.com/fab-jul/torchac

Table 1: **Discriminator Architecture**. We investigate multiple discriminator including StyleGAN (Karras et al., 2019), ProjectedGAN (Sauer et al., 2021) and UNet (Schönfeld et al., 2020).

| Discriminator | FID ↓ | KID ↓ | MS-SSIM ↑ |
|---|---|---|---|
| None | 26.1 | 1.2×−2 | 15.3 |
| StyleGAN | **3.57** | **4.8×10−4** | 13.3 |
| ProjectedGAN | 3.69 | 6.6×10−4 | **14.1** |
| UNet | 3.87 | 6.7×10−4 | 13.9 |

Table 2: **Different MIM masking policies.** PQ-MIM with quincunx pattern with 5 steps reduces the bpp significantly (27%). Additionally, PQ-MIM is orders of magnitude cheaper in terms of FLOPs compared to an autoregressive raster order masking pattern.

| Masking policy | #steps | bpp | MACs/Pixel (M) |
|---|---|---|---|
| Marginal baseline | 1 | 0.512 | 0.69 |
| Raster order | $T$ | OOM | $24.4×10^3$ |
| Quincunx | 5 | 0.373 | 6.66 |

**Datasets.**     We train our models using ImageNet (Deng et al., 2009). For data augmentation, we apply random resized cropping to 256×256 images and horizontal flipping. For evaluation and comparison to prior work, we use KODAK (Kodak, 1993) and TECNICK (Asuni & Giachetti, 2014) datasets for PSNR and MS-SSIM. Moreover, we compute the perceptual metrics (FID (Heusel et al., 2017), KID (Bińkowski et al., 2018)) for perceptually trained models using the CLIC 2020 test-set (Toderici et al., 2020) (428 images) using the same patch cropping scheme detailed by Mentzer et al. (2020).

**Baselines.**     We compare to several existing neural compression baselines: the scale hyperprior model (Ballé et al., 2018), mean & scale hyperprior (Minnen et al., 2018), GMM hyperprior (Cheng et al., 2020), and HiFiC (Mentzer et al., 2020). Among the non-neural codecs, we compare to the popular BPG (Bellard), WebP, and JPEG2000.

## 4.2   Main experimental results

**Comparison to existing (neural) codecs.** We compare PQ-MIM to other approaches across a wide range of bitrates in Figure 5[2]. Note that in our evaluation we consider bitrates that are an order of magnitude lower than what is typically studied in the literature: most previous studies were limited to 0.1 bpp and above, see *e.g* (Ballé et al., 2018; Cheng et al., 2020; Minnen et al., 2018). The extremely low bitrates we consider make it possible to transmit a 256×256 image in an SMS or a tweet (280 characters)[3] as shown in Figure 7. PQ-MIM achieves a strong and competitive performance for both KODAK and TECNICK datasets, outperforming all prior neural and standard codecs with the exception of GMM hyperprior (Cheng et al., 2020). We observe that PQ-MIM is particularly strong for low rates, making it a good fit for extreme compression scenarios. Moreover, PQ-MIM exhibits a particularly strong MS-SSIM performance which was designed to model the human visual contrast perception (Wang et al., 2004; 2003).

**Perceptual metrics comparison.**     In Figure 6, we compare PQ-MIM to HiFiC (Mentzer et al., 2020) in perceptual quality measures like FID and KID as well as MS-SSIM. HiFiC is based on the mean & scale hyperprior model (Minnen et al., 2018), but adds adversarially trained discriminator model to improve the perceptual quality of the image reconstructions. PQ-MIM, with perceptual training, achieves a strong performance for all reported metrics, outperforming HiFiC for all operating points.

## 4.3   Analysis and Ablations

**Model size and architecture.** In Figure 8, we analyze the effect of using XCiT of different capacities for the autoencoder with respect to rate-distortion trade-off. We observe that the performance improves with higher capacity autoencoders, but their is a diminishing return with further increase in capacity. For all our experiments we use an XCiT-L6 since it achieves the best performance.

**Masking patterns.**     In Table 2, we compare to predicting tokens one-by-one autoregressively in a raster-scan order, the same pattern used in VQ-VAE based generative image models such as DALL-E (Ramesh et al., 2021) and VQ-GAN (Esser et al., 2021). In contrast to PQ-MIM, raster-scan models require causal attention, which makes XCiT not a good fit. We use a standard ViT model instead. However, due to the quadratic complexity of ViT and the high resolution of images typically used for evaluation of compression

---

[2]All results for the baselines are reported using the authors official repositories
[3]For example, a bitrate of 0.03 yields $256 × 256 × 0.03/8 ≈ 246$ bytes for a 256×256 image.

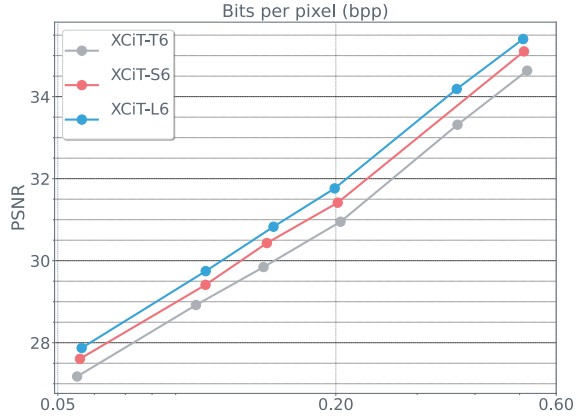
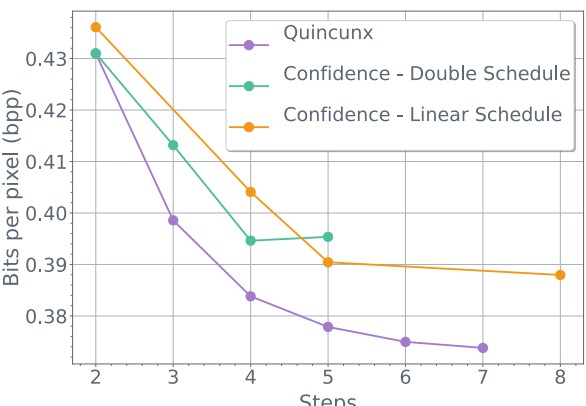

Figure 8: **Autoencoder capacity.** The RD performance of different encoder/decoder capacities. We use the same model trunk for (i) the encoder; (ii) the PQ-MIM entropy model ; and (iii) the decoder. Increasing the model size from a XCiT-T6 model (3.5M params) to XCiT-L6 (47M params) increases the performance by typically +0.8dB.

Figure 9: **MIM number of steps.** In addition to quincunx, we explore using the prediction confidence as the patch masking policy following MaskGiT (Chang et al., 2022). We test a linear and doubling schedules. Quincunx provides a higher rate saving, even compared to confidence policy with a longer schedule.

method (e.g. TECNICK), our autoregressive variant consistently exceeded the memory limits, even when using A100 GPUs with 40GB memory. Moreover, raster scan fully-autoregressive models results in extremely expensive FLOP count since it needs $T$ separate forward passes per image. On the other hand, our stage-wise MIM with quincunx pattern requires 4 evaluations, and does not scale with the image resolution as is the case for raster, making it a more practical solution.

**Number of prediction stages.** We compare the quincunx masking pattern with masking based on confidence score following MaskGiT (Chang et al., 2022) patch selection procedure for image generation. In the latter case, at a given step, we pick the patches to transmit dynamically based on their confidence score. The confidence is defined as the maximum across the probabilities assigned over the vocabulary by the model.[4] For the quincunx and confidence based masking policy we use the same 5-stage scheme, in which the number of tokens in subsequent groups doubles in size. We ablate the number of prediction stages $S$ for the quincunx and confidence-based sampling. For the latter we consider two options: (i) a linear scheme where each group of tokens contains (approximately) the same number of tokens; and (ii) a doubling scheme where each subsequent group of tokens is double the size of the previous group, as is also used for the quincunx pattern. Every point on the curves in Figure 9 corresponds to the bitrate when encoding/decoding with a given number of steps. For example, $S = 2$ steps means we have only two steps each encoding/decoding 50% of the patches. For the 3-steps doubling schedule the groups have sizes of 1/4, 1/4, 1/2, and so on. For all three patterns the bitrate monotonically decreases with the number of steps, as more tokens can be predicted from larger contexts.

On the one hand, using confidence-based masking patterns, doubling and linear schemes lead to mostly comparable bitrates for the same number of steps, with further improvement for linear with more steps however with diminishing return. On the other hand, quincunx provides a stronger reduction ratio with strictly 5 steps, even when compared with confidence-based masking with higher number of steps.

**Predicting missing patches.** When encoding/decoding the discrete image representation **q**, we reduce the bitrate with our models by their ability to predict the remaining tokens given those of preceding stages. To illustrate the predictive ability of our model, we consider an experiment where we remove a subset of the tokens (sampled randomly), and use our model to fill in the missing patches by conditioning on the observed ones. We then use the PQ-VAE decoder to decode the discrete latent codes including the filled in ones. In Figure 10 we show the results obtained when removing 10%, 20%, 30% and 50% of the patches. Our MIM is able to model redundancies among the patches, which corroborates our findings of its compression ability.

---

[4]Note that for decoding the same confidence score can be used to identify the group of tokens to decode.

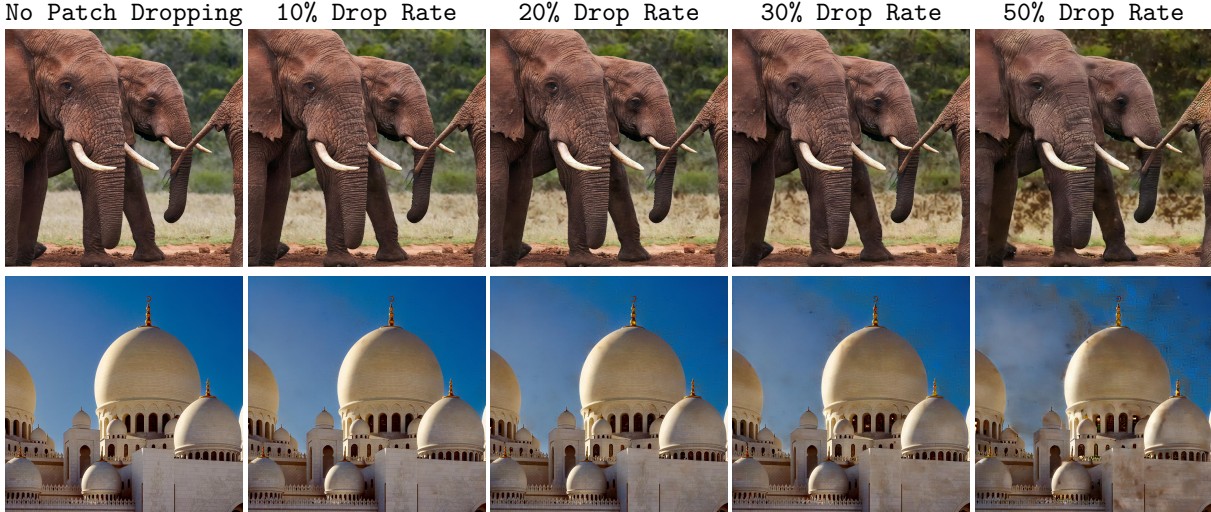

Figure 10: **PQ-MIM can operate on a partial set of transmitted tokens.** Since PQ-MIM is compatible with generation, the conditional entropy model can be repurposed to predict the PQ codes for missing parts in an image. We show results for different dropping rates of transmitted tokens. PQ-MIM exhibits strong inpainting abilities. Even for the extreme case where half of the image patches are dropped, PQ-MIM can still retain a large percentage of the original image structure and details.

### 4.4 Limitations

Image compression, and compression of visual data in general, is an important technology to scale the distribution of visual data. This is ever more important to cope with the growing quantity of visual data that is streamed in the form of video and for augmented and virtual reality applications. Compression is also critical to allow users with low-bandwidth connections to benefit from applications relying on sharing of image, video, or virtual reality data.

**Caveats of learned neural compression models: biases and performance.** As with any machine learning model, potential biases in the training data may be transferred into the model via training. In our setting, this can affect the autoencoding reconstruction abilities for content under-represented in the training data, as well as the compression abilities of the model for such data. Such biases should be assessed before deployment of the model. Beyond rate-distortion trade-offs, important evaluation dimensions include the energy and latency performance of compression models. Current neural compression methods, including ours, need to be further optimized to be competitive with existing codecs on these criteria.

**Specific limitations of our approach.** In our work, we specifically focus on a high compression regime, with compression rate lower than 0.6 bit per pixel. This is a favorable case for our approach: it benefits from the generative model capability inherited from the VQ-VAE. Compared to VQ, PQ allows for higher rates, but it becomes comparatively less effective when increasing the number of subquantizers. As one can deduce from the slopes of the rate-distortion curves in Fig. 5, SQ methods like scale hyperprior exhibits a relatively stronger performance for the high end of bit-rates.

## 5  Conclusion

In this paper, we have revisited vector quantization for neural image compression. We introduced a product-quantization variants of VQ-VAE and shown that it has a better scaling properties in terms of bit-rate. Additionally, we introduced a novel conditional entropy model based on masked image modeling. We have shown that combined with the qunicunx partitioning pattern, PQ-MIM provides strong reduction in bit-rate. PQ-MIM exhibits a competitive performance for PSNR and MS-SSIM metric compared to strong neural compression baselines. Furthermore, when we train PQ-MIM using perceptual losses, it provides a strong performance on multiple metrics (e.g. FID, KID and MS-SSIM) compared the strong baseline of HiFiC. Finally, we have shown that PQ-MIM can operate in a hybrid compression/generation mode where it can fill the gaps for non-transmitted patches.

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
