# OpenReview forum: "Image Compression with Product Quantized Masked Image Modeling"
_TMLR — Accepted by TMLR_

### Review · Reviewer_yFdz · 2023-01-09

**Summary Of Contributions:**

First, the authors propose a generative model for image compression, representing a middle ground between standard variational autoencoders (VAE) and vector quantized VAEs (VQ-VAEs). In particular, the authors propose to use product quantization (PQ). PQ partitions the embedding space into a set of subspaces and trains a separate vector quantizer for each subspace. Thus, one can use much smaller codebooks for each vector quantizer while retaining good flexibility of the model.

Second, the authors propose a hierarchical prior model for the product-quantized representations using a masked image model (MIM). Hence they call their model PQ-MIM.

The authors conduct usual image compression experiments and report results on the standard datasets Tecnick and Kodak. They also perform ablation studies on the model size, the masking patterns and the masking process, and imputing latents using their predictive model.


**Audience:**

Yes

**Broader Impact Concerns:**

-

**Claims And Evidence:**

Yes

**Requested Changes:**

- [Necessary] If there is a connection with hierarchical VAEs, the authors need to discuss this connection.
- [Necessary] Some empirical studies on how PQ performs on its own relative to VQ.
- [Optional] The first paragraph of the introduction is a bit too grandiose and doesn't contribute to the paper's content. Hence it should be heavily shortened or removed altogether.

**Strengths And Weaknesses:**

# Strengths
The paper is well-written and easy to follow, with only a few minor mistakes that can be easily fixed.

The authors study a relevant problem, and their proposed solution is sound. Furthermore, I appreciated that the authors put a good amount of thought into the time complexity of the method to ensure its scalability, e.g. via their choice of transformer architecture.

The authors conduct a reasonable amount of ablation studies to confirm the validity of their modelling choices. And they show that their method with their proposed architecture performs at least on par and sometimes clearly better than the state-of-the-art on the standard set of distortion and perceptual metrics. The authors also demonstrate that their method scales gracefully to the very small bitrate regime and that their learnt prior model is strong enough to impute a large percentage of missing latents.

# Weaknesses
I have a couple of prominent points to clarify, alongside a few more minor ones.

First, the relation of PQ-MIM to hierarchical VAEs is unclear. The way Eqs (2) and (3) define the prior structure look identical to the generative model of a particular hierarchical VAE with $S$ stochastic layers where there are an equal number of latents on each layer. Hence, the power of PQ-MIM, in terms of coding efficiency and generative / imputation ability, could be explained by the power of hierarchical VAEs. It also makes the distinction the authors make between PQ-MIM and hierarchical VAEs seem artificial. Could the authors comment on this?

Given the above, it is unclear to me why any partition of the latent space into equal-sized sets should be special so long as each set in the partition reasonably covers the whole latent space (as opposed to each set in the partition being a contiguous "patch", like a set of squares or rectangles). In particular, I don't see what causes the discrepancy reported in Figure 9 between different partitioning techniques. Uniformly randomly partitioning the latent space into $S$ sets would also be a good baseline.

Further, it is unclear how the two proposed improvements of PQ and the prior model perform independently. As I understand, the number of subvectors $M$ and the number of prior partitions $S$ are independent. For example, it would be important to know how much flexibility is lost using PQ instead of full VQ. For example, how does the performance of a VQ-VAE with a codebook of size 16 x 16 x 16 for each latent compare to a PQ-VAE with $M = 3$ subvectors per latent and a codebook of size 16?

Some more minor points are:
 - In the introduction, the authors pose the following question: "How to define a vector quantizer offering a range of rate-distortion operating points?" - I am not sure what the authors meant here. This sentence reads as if the authors were proposing a variable-rate compression method. While dropping some of the transmitted tokens, as done in Section 4.3, could be conceived as doing this, if the authors want to claim variable rate compression as a contribution, they need to examine the performance of their method more thoroughly.
 - Roughly, how do the computational costs of the model the authors used to generate results reported in Figure 5 compare with the cost of other methods?
 - In the contributions, the authors claim: "This simple PQ-VAE variant offers a strong and scalable rate-distortion trade-off" - I get that a strong here means that the method gives a good performance, but what is meant by a "scalable rate-distortion trade-off"?
 - "In particular we consider the "quincunx" regular grid pattern, where in each stage we double the number of tokens to predict, see Figure 4 for an illustration" - This is not what Figure 4 shows. Figure 4 shows that at each stage the same number of new tokens are revealed.
 - "Without loss of generality, we choose all neural network models to be identical" - What generality is not lost?
In section 3.1: How was Figure 3 generated? At least provide a pointer to the description of how it was generated. Perhaps include some description in the caption.

---

> ### Author Response · Authors · 2023-02-09
> **Author's response to Reviewer yFdz**
>
> We thank the reviewer for acknowledging the paper's contributions and strengths.
>
> - **If there is a connection with hierarchical VAEs, the authors need to discuss this connection.**
> 	+ The hierarchical VQ-VAE model aims at improving the generations/reconstructions quality by exploiting *multi-scale* representation and this is achieved by architectural modifications to capitalize on the innate hierarchy in the conv-net-based encoder to extract multi-scale features. Each scale in the hierarchical features is then vector-quantized. The reviewer mentions a similarity between hierarchical VQ-VAE and PQ-MIM if the hierarchical features are designed to have the same scale. That being said, there is a similarity in that both PQ-MIM and hierarchical VAEs rely on a stage-wise prior which models certain latents as conditionally independent given "previous" latents. We do want to highlight crucial differences between the approaches. (1) The PQ component in PQ-MIM operates on a single set of latents and it follows a rich literature on product quantization that precedes hierarchical VQ-VAE. Hierarchical VQ-VAE is designed to operate on different sets of latents each with its associated scale, therefore reusing the same features for both scales does not stick to the motivation and design choices of hierarchical VQ-VAE. (2) For PQ-MIM, it is trivial to expand the number of PQ sub-vectors by simply expanding the latent dimension or reducing the dimension of each sub-vector, to the extreme where we approach SQ. However, for hierarchical VQ-VAE it is not clear how to expand the number of levels/scales beyond a few scales.
>
> - **Some empirical studies on how PQ performs on its own relative to VQ.**
> 	+ We thank the reviewer for their suggestion. We have included in the revision Figure B2 in which we compare the PSNR obtained using VQ and PQ, independently of the MIM component, using only the histogram of frequencies for the codebook entries as the CDF for the AE/AD.
>
> - **The first paragraph of the introduction is a bit too grandiose and doesn't contribute to the paper's content. Hence it should be heavily shortened or removed altogether.**
> 	+ We thank the reviewer for their suggestions. We have simplified the opening paragraphs in the revision.
>
> - **Uniformly randomly partitioning the latent space into sets would also be a good baseline.**
> 	+ In preliminary experiments, we did not observe significant differences in bpp obtained using random and confidence-based partitioning. We retained the confidence-based scheme for brevity, and because it does not require a scheme to communicate the group partitioning scheme as in the random partitioning case.
>
> - **How was Figure 3 generated? At least provide a pointer to the description of how it was generated. Perhaps include some description in the caption.**
> 	+ We have added some implementation details about the PQ and VQ models to the caption of Figure 3 based on the reviewer's feedback.
>
> - **"This simple PQ-VAE variant offers a strong and scalable rate-distortion trade-off" - I get that a strong here means that the method gives a good performance, but what is meant by a "scalable rate-distortion trade-off"?**
> 	+ By scalable we refer to the fact that we can trivially scale the bit-rate of PQ models by increasing the number of sub-vectors while keeping the same optimization difficulty by restricting the codebook size. On the other hand, for VQ increasing the bit-rate can be challenging since the codebook size needs to grow exponentially for a linear gain in bit-rate and therefore the optimization problem becomes harder rapidly.
>
> - **"How to define a vector quantizer offering a range of rate-distortion operating points?" - I am not sure what the authors meant here. This sentence reads as if the authors were proposing a variable-rate compression method.**
> 	+ We do not propose a variable rate compression method and our method requires a separate model for each specific R-D tradeoff, which is similar to the majority of the literature. The study of variable rate compression methods is a very important research direction, however, we do not attempt to tackle it in this work.

---

> > ### Comment · Reviewer_yFdz · 2023-02-12
> > **Response to the authors**
> >
> > I thank the authors for their response. I have three outstanding worries:
> >  - I wonder if we are approaching the problem from two separate perspectives. The authors' response to my question regarding how PQ-MIM differs from hierarchical VQ-VAEs doesn't actually demonstrate that they are different. It seems to me that the authors are claiming that PQ-MIM is different from the specific VQ-VAE architecture given in [1], which I agree with. But at a conceptual level, i.e. conditioning on vector-valued latents to predict another set of latents, I still don't see how they differ.
> >  - Why did the authors put their comparison results with more recent methods in the appendix in Figure B3? This comparison plot, or at least a trimmed version, should be in the main text.
> >  - The authors haven't responded to my concern regarding Figure 4, and I worry that the statement I quote in my review is false. Could the authors respond, please?
> >
> > I would also like to note that even though it is not a direct requirement for acceptance in TMLR, it would be nice if the authors could motivate their method better. Reviewer VDgp asked what the justification was for using VQ/PQ for compression. The authors responded by saying that their contribution is that they demonstrate that VQ/ PQ can be competitive with SOTA scalar quantization methods. However, it would be good if the authors could point out what more concrete benefits they see in bringing methods from generative modelling closer to methods for data compression.
> >
> > ## References
> > [1] Van Den Oord, A., & Vinyals, O. (2017). Neural discrete representation learning. Advances in neural information processing systems, 30.

---

> > > ### Author Response · Authors · 2023-02-17
> > > **Response to reviewer yFdz**
> > >
> > > - **Hierarchical VQ-VAE**
> > >
> > > We want to point out an important difference. If we ignore the MIM components which only acts as an entropy model and does not impact the reconstructions and their quality, the PQ-VAE model will be a better counterpart to the hierarchical VQ-VAE. In that case, product quantization does not rely on any conditioning on other vector values, each sub-vector is rather quantized independently. As we mentioned the initial response, PQ-VAE has more flexibility to scale the number of sub-vectors trivially while it is not as clear for hierarchical VQ-VAE since it is inherently hierarchical.
> > >
> > > - **Why did the authors put their comparison results with more recent methods in the appendix in Figure B3?**
> > >
> > > The only reason is not the break the symmetry of Figure 5 since the reported methods didn't provide MS-SSIM trained checkpoints. The new results does not change the conclusions of the work since the same behaviour remains where PQ-MIM is stronger for lower rates while the reported methods matches/surpasses PQ-MIM for the highest reported rates,
> > >
> > > - **The authors haven't responded to my concern regarding Figure 4, and I worry that the statement I quote in my review is false. Could the authors respond, please?**
> > >
> > > We apologise for missing this point. In Figure 4, each new step the tokens the method is tasked to predict are doubled (same for new revealed tokens) which matches our the method we describe in the paper.
> > >
> > > - **It would be good if the authors could point out what more concrete benefits they see in bringing methods from generative modelling closer to methods for data compression.**
> > >
> > > As detailed in the contributions in the paper, we qualitatively show that PQ-MIM is capable of operating in a hybrid mode, between generative and compression, without requiring further training and finetuning. This allows for higher resilience to corrupted or missing signal where our model can fill-in the missing information.

---

> > > > ### Comment · Reviewer_yFdz · 2023-02-17
> > > > **Response to the authors**
> > > >
> > > > I thank the authors for their response. They made me realize that I was conflating the entropy model with the actual architecture, so hierarchical VQ-VAEs are not quite comparable to the contents of the paper. I also realize that my comment regarding Figure 4 wasn't correct. What confused me is that in the second pane, the same number of tokens was revealed as in the first pane, and somehow that caused enough cognitive dissonance that I didn't notice that from the third pane onwards, the number of tokens does indeed double. Is the fact that there is no doubling only in the second pane a mistake, or is this how the pattern works and is just an unnoteworthy implementation detail?
> > > >
> > > > Otherwise, the authors have addressed my concerns, and I recommend acceptance.

---

> > > > > ### Author Response · Authors · 2023-02-17
> > > > > **Response to reviewer yFdz**
> > > > >
> > > > > Concerning Figure 4, the reviewer's assumptions is correct. It is indeed an implementation detail specific to the first step only where moving from the seed observed tokens (obtained using the marginal entropy) to the next step the tokens are not doubled, but for all following steps the tokens are doubled.

---

### Review · Reviewer_GQkS · 2023-01-24

**Summary Of Contributions:**

The authors propose a novel deep image compression model that combines several novel techniques proposed in recent years. The proposed method groups input pixels (tokens) into a quincunx pattern. Pixels in the same group are encoded / decoded parallelly. A progressive decoder is then applied on top of groups instead of pixels, which accelerates decoding speed. The model is trained to minimize the MSE or perceptual loss. The authors demonstrate the performance improvement of the proposed method against popular codecs and deep learning based image compression baseline methods.


**Audience:**

Yes

**Claims And Evidence:**

Yes

**Requested Changes:**

The deep image compression baseline models compared in the experiments are not update-to-date. Only methods proposed in 2018-2020 are compared. Please consider to compare to more recent works, especially 2021-2022 works.

One of the key arguments of this work is to accelerate the encoding /decoding speed via parallalization with token groups. However, there is no speed comparison in the experiments. Please append speed comparison experiment.

The multi-stage approach significantly improves bit rate but might incur with slower decoding speed. Would be better to compare the speed v.s. number of stages v.s. bit rate together.

It seems that, more stages always improve bit rate. Let us image that the number of stages is being pushed into infinite. What would we expect? Is there a break point that the performance becomes worse agaist more steps?

It is not very clear to me that at the last step, how many tokens will be used in encoding / decoding. From Eq. (3), it seems all decoded tokens will be used as input. If this is true, wow to deal with increasing number of tokens in multi-step decoding?

**Strengths And Weaknesses:**

The paper is over all well-writen and easy to follow. High-level intuitions are well explained with figures. Numerical experiments are provided with sufficient details. Experiments are well organized.

The multi-stage encoding / decoding in an interesting idea. From the numerical experiments, more stages give better compression rate.

The numerical experiments show that the proposed method outperforms conventional JPEG and BPG as well as some deep learning baselines.

---

> ### Author Response · Authors · 2023-02-09
> **Author's response to Reviewer GQkS**
>
>
> We thank the reviewer for their comments and feedback, we address their concerns and questions below.
>
> - **Comparison to more recent works**
> 	+ We thank the reviewer for their suggestions. We have added more comparisons to recent methods in Figure B3 in the revision.
>
> - **Would be better to compare the speed v.s. # stages v.s. bit rate together.**
> 	+ We thank the reviewer for their suggestion. We have added Figure B1 in the revision which plots the bpp and time(ms) as a function for decoding steps.
>
> - **One of the key arguments of this work is to accelerate the encoding /decoding speed via parallelization with token groups. However, there is no speed comparison in the experiments. Please append the speed comparison experiment.**
> 	+ We have added Table A1 in the revision that compares Quincunx with 5 steps to the naive baseline of decoding one token per step (similar to autoregressive methods) and show a very significant speed up using our strategy.
>
> - **Let us image that the number of stages is being pushed into infinite. What would we expect?**
> 	+ Adding more groups will allow the model to leverage more dependencies among the tokens and generally reduce the bitrate.
> 	+ Consider increasing the nr of groups by one. The tokens in the original first group are now split across two equal-sized groups, say 1a and 1b. Previously, the entire original first group was modeled by the marginal distribution over the vocabulary, and now only 1a will be modeled by the marginal. As long as the MIM module has enough capacity to learn the marginal distribution, it can always resort to the marginal to model the newly introduced group 1b, or do better by leveraging the dependencies across groups 1a and 1b.
> 	+ Ultimately, when the number of groups increased, there will be only one token left per group, a fully autoregressive model is recovered, which yields prohibitively slow decoding.
>
> - **It is not very clear to me that at the last step, how many tokens will be used in encoding / decoding, How to deal with increasing number of tokens in multi-step decoding?**
> 	+ Each step is equivalent to a single forward to our transformer MIM module where all observed tokens are processed in parallel, and all remaining tokens are predicted. Therefore, we need strictly S forward for S quincunx steps and the number of forward doesn't scale with the number of tokens or image size in contrast to autoregressive methods. Therefore, for the last step where half of the tokens are observed and half are masked, the masked positions will be replaced by a "MASK" embedding vector and all tokens will be passed through the MIM transformer via a single forward pass.

---

### Review · Reviewer_VDgp · 2023-01-26

**Summary Of Contributions:**

This paper proposes a new quantization scheme and entropy model for neural image compression, drawing inspiration from recent work in image generation models. Specifically, the idea of vector quantization (from VQ-VAE) was adapted into product quantization for better scalability, and an autoregressive entropy model based on a quincunx pattern was proposed. The combined architecture yields competitive compression performance with existing neural image compression methods.



**Audience:**

Yes

**Broader Impact Concerns:**

None.

**Claims And Evidence:**

No

**Requested Changes:**

**Main changes** (also described in the "weaknesses" above):
1. A better discussion / motivation for VQ/PQ v.s. scalar quantization, even if the result is negative. Classical results in quantization/compression indicate that at high bit-rates, uniform scalar quantization is close to optimal (see Ziv, 1985), which seems to speak against VQ/PQ. Examining this result and understanding to what extent this holds in non-linear transform coding would add to the scientific value of this paper.
2. A discussion on the related work of checkerboard-patterned entropy model (He et. al., 2021), which would help better place this work in the literature.
3. A more rigorous R-D-P comparison to HiFiC, in order to justify the claim that the method outperforms HiFiC. One valid procedure could be fixing one of the three axes and comparing the remaining two.

Ziv 1985. "On Universal Quantization". IEEE TRANSACTIONS ON INFORMATION THEORY.


Minor typo:
p. 3, "with a prefix that encoding the conditioning text" -> "with a prefix that *encodes* the conditioning text"

**Strengths And Weaknesses:**

**Strengths**:

The paper introduces several new ideas for improving the entropy modeling in non-linear transform coding, and does a fairly comprehensive job evaluating different architectural choices of the proposed entropy model. The resulting method offers competitive compression performance compared to SOTA methods, as well as visually pleasing progressive reconstructions.

**Weaknesses**:

It's a bit difficult to understand the effect of the various aspects of the proposed approach and how they are justified.
Specifically, there is a lack of clear motivation for preferring vector/product quantization instead of the baseline scalar quantization approach -- other than the appeal to the success of VQ-VAE on image modeling. Isn't it possible to still apply the transformer-based entropy model with scalar-quantized latents (e.g., as in Mentzer et al., 2022)? It'll be helpful to understand the effect of VQ/PQ v.s. SQ separately from the newly introduced entropy model.
Similarly, the proposed Quincunx-patterned entropy model appears similar to checkerboard-patterned entropy model (He et. al., 2021), with largely the same motivation; therefore a discussion/comparison seems appropriate.
Finally, I'm not sure if the rate-distortion-realism comparison to HiFiC in Fig. 6 is done 100% correctly. My understanding is that for comparing R-D-P tradeoffs, the appropriate procedure is to fix one of the axes and compare the remaining two; e.g., training a couple of models for low/mid/high rate regimes for the proposed method, and compare the D-P tradeoff with HiFiC at *matching bit-rates* (as done in the HiFiC paper).

References:

Mentzer et al., 2022. "VCT: A Video Compression Transformer". NeurIPS 2022.

He et al., 2021. "Checkerboard Context Model for Efficient Learned Image Compression". CVPR 2021.

---

> ### Author Response · Authors · 2023-02-09
> **Author's response to Reviewer VDgp**
>
> We thank the reviewer for their valuable feedback and suggestions. We detail below the clarifications and answers that we hope can address their concerns.
>
> - **VQ/PQ comparison to SQ**
> 	+ First, we want to clarify that our main motivation is to bridge the gap between generation and compression methodology and therefore we choose VQ/PQ. Since the vast majority of neural compression papers rely on SQ and it is very established in the literature, we consider it a contribution that we show that we can obtain competitive performance with PQ or VQ. PQ is a generalization of SQ and VQ, and allows compromises between both.
> 	+ VQ/PQ is more compatible with MIM since the model is tasked to conditionally predict a single discrete value in the case of VQ or a handful for PQ (e.g. no more than 6 in our setup) for each token. If we apply our method to 	SQ, we need to predict a high number of symbols for each token (typically 192), which complicates the method either in terms of parameters (if we use separate heads per symbol) or relying on the channel-wise autoregressive model with very high interence compute cost.
> 	+ Note that in the limitations section, we already mention that PQ-MIM, despite its strength for the low bit-rate regime, doesn't scale as well for very high rates compared to SQ methods, as empirically observed in the right side of RD plots in Figure 5. We clarified this point further in the revision.
>
> - **discussion on the related work of (He et. al., 2021)**
> 	+ We thank the reviewer for suggesting this additional reference. We have updated the related work section with the inclusion of a discussion of the work of He et. al. 2021. Please note that the method proposed by He et al. can be considered a special case of our Quincunx decoding with S=2 groups of tokens, and the contextual decoding convolutional model are replaced by a transformer and MIM objective. PQ-MIM generalizes the checkboard pattern to multiple steps using the quincunx pattern and we demonstrate that more steps are beneficial in Figure 9.
>
> - **R-D-P comparison to HiFiC**
> 	+ First, we want to mention that in the paper we only claim that PQ-MIM outperforms HiFiC in terms of FID and KID, which is supported by the experimental data provided in Fig. 6.
> 	+ In the paper we report two perceptual measures (KID and FID) and a distortion measure (MS-SSIM). We observe that PQ-MIM is better on three of these metrics for several bitrates. We acknowledge that there is a trade-off between perceptual quality and distortion, but our aim in the current paper is to compare to the SoTA methods in terms of perceptual quality, of which HiFiC is the most notable example. Managing the trade-off between perceptual quality and distortion is an interesting problem that has been recently studied by mutliple papers [1, 2], but it is outside the scope of the work described in the current paper.
>
> [1] Agustsson, Eirikur, et al. ""Multi-Realism Image Compression with a Conditional Generator."" arXiv preprint arXiv:2212.13824 (2022).
>
> [2] Muckley, Matthew J., et al. ""Improving Statistical Fidelity for Neural Image Compression with Implicit Local Likelihood Models."" arXiv preprint arXiv:2301.11189 (2023)."

---

> > ### Comment · Reviewer_VDgp · 2023-02-13
> > **Thanks for the rebuttal. Two remaining concerns.**
> >
> > Thank you for the response, which has addressed some of my concerns. However, there are still a few concerns I feel are inadequately addressed:
> >
> > 1. **On motivation for VQ/PQ, esp. compared to SQ**: Thanks for offering some insight on this, although it seems the main reason to prefer PQ over SQ is that SQ is less compatible with the proposed MIM entropy model, rather than inherent limitations of SQ itself.
> > The authors also suggested that PQ generalizes SQ and has "strength for the low bit-rate regime", though it "doesn't scale as well for very high rates compared to SQ". It would be great if there can be an empirical comparison b/w PQ and SQ to support these ideas, perhaps in a small-scale ablation where doing SQ with the proposed entropy model is still computationally feasible (or, if not, I suppose a standard hyper-prior entropy model could be used, while keeping the non-linear transforms the same).  Essentially something similar to the newly added Figure B2 in the appendix in response to reviewer yFdz, but for SQ v.s. PQ rather than VQ v.s. PQ.
> >
> > 2. **Comparison to HiFiC.**
> > Could the authors confirm that each of the the comparisons in Figure 6 was based on the same set of model checkpoints? i.e., the same HiFiC checkpoint was used to evaluate FID/KID and MS-SSIM; the same for PQ-MIM? If so, then this suggests the PQ-MIM result is more Pareto-optimal than HiFiC (outperforming on rate, distortion, and divergence, when measured in bpp, MS-SSIM, and FID/KID), and largely addresses my earlier concern.    Otherwise, please provide the divergence v.s. rate comparisons at matching distortion levels (or, in general, fix one of (R, D, P) and compare the other two), as otherwise the result can be completely meaningless.

---

> > > ### Author Response · Authors · 2023-02-17
> > > **Addressing remaining concerns**
> > >
> > > We thank the reviewer for their response.
> > >
> > > 1. **On motivation for VQ/PQ, esp. compared to SQ**
> > >  We have experimented with implementing SQ in our pipeline, specifically we use sub-vector dimension of 1 (scalar) and use a binary codebook per sub-vector. We have observed that training such models yields very weak performances (PSNR: 10 -> 14, Rates: 0.0625 -> 0.75). We hypothesis that it can be very challenging to attain strong performances for SQ in a simple VQ-VAE like pipeline and a more sophisticated components like that utilized by the hyperprior line of work is required to have a strong performance. When we mentioned  "doesn't scale as well for very high rates compared to SQ" we referred to state-of-the-art methods that are typically based on the hyperprior model. Our method is extremely simple and highly matches the generative pipeline which is one of the main claims of the paper that we support with our experiments.
> > >
> > > 2. **Comparison to HiFiC**
> > > We confirm that the same checkpoint was evaluated for FID/KID and MS-SSIM in Figure 6. Same for HiFiC, we report the results detailed by the HiFiC authors in their official code (https://github.com/tensorflow/compression/blob/master/models/hific/data.csv)

---

> > > > ### Comment · Reviewer_VDgp · 2023-02-28
> > > > **Thanks for the response.**
> > > >
> > > > Thanks for the additional experiment and clarifying the HiFiC comparison setup. They have addressed my remaining concerns.

---

### Comment · Action_Editors · 2023-01-30
**Discussion phase has begun**

3 reviews have been submitted and the discussion phase has begun.
Thanks a lot to all the reviewers for their efforts in reviewing this paper.

In this phase, authors can take into account the reviewers’ feedback, answer any questions, and update the manuscript accordingly.

The reviewers can continue the discussion with the authors to gather all the information needed for submitting a decision recommendation.

---

### Decision · Action_Editors · 2023-03-04

**Recommendation:** Accept with minor revision

**Comment:**

All three reviewers acknowledged the contribution of the paper and recommended its acceptance. I agree with the reviewers to recommend accepting.

In the official recommendation, one reviewer suggested that the authors polish their references; "for instance, some arXiv papers have been published in conferences or journals, and some abbreviations should be written in full format." While it is common to cite arXiv papers, it would be good to cite the conference or journal versions if they have been accepted. Additionally, the authors could consider unifying the style of the references, as currently, both abbreviated and full formats are used.

**Audience:**

This work would be of interest to researchers working in neural data compression and generative models.



**Claims And Evidence:**

The focus of this paper is on exploring the neural image compression problem, proposing a novel quantization scheme and entropy model inspired by recent developments in image generation models. The objective is to close the gap between the methodologies used for image generation and compression, while also thoroughly evaluating various architectural choices for the proposed entropy model. The resulting approach achieves competitive compression performance compared to state-of-the-art methods.